# Evaluation of Yellow Mealworm Meal as a Protein Feedstuff in the Diet of Broiler Chicks

**DOI:** 10.3390/ani10020224

**Published:** 2020-01-30

**Authors:** Usman Elahi, Jing Wang, You-biao Ma, Shu-geng Wu, Jinlong Wu, Guang-hai Qi, Hai-jun Zhang

**Affiliations:** 1Key Laboratory of Feed Biotechnology, Ministry of Agriculture and Rural Affairs, National Engineering Research Center of Biological Feed, Feed Research Institute, Chinese Academy of Agricultural Sciences, Beijing 100081, China; usmanelahi@gmx.de (U.E.); wangjing@caas.cn (J.W.); myb0514@126.com (Y.-b.M.); wushugeng@caas.cn (S.-g.W.); 2DSM (China) Animal Nutrition Center, Bazhou 065700, China; Jinlong.Wu@DSM.com

**Keywords:** *Tenebrio molitor*, insect meal, mealworm, broiler, meat quality

## Abstract

**Simple Summary:**

Poultry meat is considered to be an available and inexpensive protein source for humans. Poultry meat acceptability always depends on meat quality. As resources decrease, new protein sources are introduced to the market. Thus, insect meal has emerged as an alternative feed source for poultry diets. Insect meal achieved its status in the market due to providing superior meat quality. In this study, we explored the inclusion of dried and fresh yellow mealworm meal in poultry diets. Yellow mealworm meal is acceptable as a protein feedstuff in the broiler diet without any adverse effects on chicks’ performance, and resulted in improved and comparable meat quality.

**Abstract:**

Yellow mealworm meal (MWM) as a protein feedstuff in the broiler diet was investigated based on the growth performance, hematological characteristics, carcass, and meat quality of broiler chicks. A total of 700 one-day-old Ross 308 male broiler chicks were assigned to five dietary MWM treatments containing 0%, 2%, 4%, and 8% dried MWM or 10.48% fresh mealworm (corresponding to 4% dried MWM). For each treatment, there were seven pens with 20 chicks each. The nutritional profile of dried MWM is comparable to all conventional protein feedstuffs. MWM significantly increased BW and ADG (linear and quadratic, *p* < 0.05), and FCR was best at 4% MWM inclusion level (quadratic, *p* < 0.10) for broiler chicks during the starter phase. The predicted MWM levels for optimal starter BW and ADG were 4.13% and 3.84%. Hematological characteristics of broiler chicks fed on the MWM diet did not differ or showed small change within the physiological range. A fresh 10.48% mealworm diet significantly reduced the blood LZM for the grower. Broiler Chicks fed on fresh 10.48% mealworm had a significantly reduced abdominal fat percentage compared to the 4% dried MWM counterparts. MWM did not significantly affect meat quality. Taken together, MWM inclusion in broiler diet is acceptable as a protein feedstuff, and a 4% level could stimulate early growth in the starter phase.

## 1. Introduction

Soybean and fish meal are conventional protein sources. Soybean is the most common vegetable protein source in feed formulation for broiler chicks and laying hens due to its protein quality and quantity and appropriate amino acid profile [1]; however, fishmeal is the most significant standard protein source of animal feed in some developing countries [2]. Currently, soybean and fishmeal accessibility and price are the leading problem. In connection with the extensive use of soybean and fishmeal, severe environmental issues must also be regarded. Increased soy cultivation leads to deforestation, elevated water consumption, and chemical usage [3]. Meanwhile, fishmeal qualitatively and quantitatively depends on fish catch, which is expected to decline in the coming years owing to the danger of depletion of marine assets [1].

Insects have been suggested as high quality, efficient, sustainable alternative protein sources. One alternative deemed to decrease the price of protein supplements in poultry feed is protein enriched insects. Moreover, chitin, lauric acid, and antimicrobial peptides in insect meal promote chicken’s health [4]. Insects can be used in dried or fresh form in poultry diets [5]. Some poultry nutritionists insist that the live (fresh) insects are the natural way for poultry. Yellow mealworm (*Tenebrio molitor* L.) is an important insect with the potential for use as a protein source to replace soybean meal and fishmeal in poultry diets [6]. Mealworm meal (MWM) is rich in protein, fat, energy, and fatty acids; thus, it can be successfully used as feedstuff in poultry diets [6]. However, the high water content in fresh insects could be sensitive to degradation and microbial activity; moreover, a Millard reaction could cause coloring, flavoring, and redox compounds formation due to the high water content [7]. In addition, the presence of digestive enzymes in insects could also influence protein properties after grinding [8]. Thus, drying insects via heating seems to be suitable for feed production. Also, proper processing of insects makes it gluten-free [9]. Although, heat treatment is beneficial from a safety point of view, denaturation and a Millard reaction could affect the solubility and availability of essential amino acids. Still, there is much controversy about the suitable (dried or fresh) form of insect utilization in poultry diets.

MWM contains high amounts of crude protein (CP) 25–60% and fat 15–40%. The recently reported values for CP are 47% dry matter (DM) [10], 53.8% (DM) [7], 46.44% [11], 54.4% [3], 53% [12], 45.83% [13], and 51.9% [14,15]. Protein from MWM is, in particular, high in essential amino acids. The essential amino acid index (EAAI) of MWM is higher than soybean meal and comparable or even higher than fishmeal.

Fresh MWM has been successfully used in fish diets [16]; however, fresh MWM usage for poultry diets is not reported. Broiler chicks fed on fresh housefly maggots had higher BW, ADG, and ADFI [17]. Moreover, supplementation with live maggots improved the growth rate, clutch size, egg weight, hatchability, and chick weight of free-range chickens [18] and reduced the fearfulness in young pullets [19]. In recent studies, significantly improved growth performance, blood profile, immune system, villous height, crypt depth, carcass, and meat quality was observed in broiler chickens [10,12,13,15,20], Japanese quails [11], and barbary partridges [14] fed on the diet containing dried MWM. Insect meal in broiler chicks’ diet can improve the meat quality, which is due to the functional ingredients in the insect [21]. Moreover, diets containing full-fat MWM have the potential to reduce Bacteroides–Prevotella cluster and *Clostridium perfringens* in broiler chicks [22].

Based on the literature, MWM may serve as a promising protein feedstuff for poultry. Thus, the current study was conducted:(i)To investigate the potential of yellow MWM as a protein source for poultry diets,(ii)To evaluate the effect of dried MWM diet and fresh mealworm diet on the growth performance, hematological characteristics, carcass composition, and meat quality.

## 2. Materials and Methods 

All experimental procedures were reviewed and approved by the Animal Care and Use Committee of the Feed Research Institute of the Chinese Academy of Agricultural Sciences (FRI-CAAS20181112).

### 2.1. Mealworm Processing

Two hundred and fifty kg of fresh yellow mealworm larvae were purchased from Hao Cheng Mealworms Inc. Qinhuangdao, China). One hundred and forty kg fresh yellow mealworm larvae were oven-dried (Tianjin taist Instrument Co., Ltd, Tianjin, China) at 50 °C for three days and ground as described by [23]. Meanwhile, one hundred kg of fresh mealworm was pulped with a fruit-pulping machine (Midea Electronics, Foshan, China) and stored at −18 °C in the deep freezer (Haier Electronics, Qingdao China) until further utilization. Nutritional compositions of fresh mealworm, dried MWM, and feed samples were analyzed for dry matter (drying in oven at 103 °C for 8 h), CP (N × 6.25) (ID: 976.05 [24]), amino acids (ID: 994.12 [25]), fat (ID: 920.39 [24]), calcium (ID: 927.02 [24]), phosphorus (ID: 964.06 [24]), and gross energy by the bomb calorimeter method.

### 2.2. Birds and Diets

Seven hundred one-day-old Ross 308 male broiler chicks (individual BW~42g) vaccinated against Newcastle disease and Infectious Bronchitis, were randomly allotted to the five dietary treatments (7 pens/treatment and 20 birds/pen) and reared on the floor with wood shavings as a bedding material. Each pen (0.9 × 1.7 m^2^) was provided with individual feeder and drinker. The management of the broiler chicks was performed according to the guidelines for Ross broiler chicks. Feed (cold pellet form) and water were provided ad libitum to broiler chicks and a 243L:1D light regime was maintained. The temperature in the chicken house was maintained at 33 °C for the first three days and then reduced by 3 °C each consecutive week until 28 °C. Four diets containing dried insect meal (0%, 2%, 4%, and 8%) and one diet containing fresh insect meal (10.48%) were formulated. The feed formulation for the diet containing fresh (10.48%) mealworm corresponded to that of dried 4% MWM excepted the mealworm type and inclusion. The control diet was formulated according to the [26,27] recommendations. The dry matter contents of fresh insects and dried insect meal were 36.41% and 95.40% respectively. The frozen mealworm was taken from the freezer and allowed to thaw out at room temperature. The thawed pulp was added into the mixture slowly and mixed with other premixed ingredients. The diet for fresh mealworm was mixed in two steps, firstly with other ingredients in the absence of fresh meal, and then with the fresh meal. The prepared powder diet was then pelleted by a flat die extrusion granulator as other diets. The ingredients and nutritional profile of the diets for starter (0–21d) and grower (21–42d) phases are presented in Table 1.

### 2.3. Growth Performance

Bodyweight (BW), feed offered (FO), and feed residue (FR) were documented for each pen on 0, 21, and 42 d of age. Mortality was documented daily for FI correction. Feed conversion ratio (FCR), average daily feed intake (ADFI), and average daily gain (ADG) were calculated by using the documented data of BW, FO, FR, and mortality.

ADG = Final BW − Initial BW/Age in days

FCR = FI/ADG

### 2.4. Hematological Characteristics

One chick was randomly selected from each pen for blood sampling at 21 and 42 d of age. Three m of blood was sampled from the wing vein and kept in heparin-lithium-treated tubes. The blood was then centrifuged at 1800 g for 10 min, and the plasma was stored in a 1.5 mL Eppendorf tube at −30 °C until analysis. The activity of total superoxide dismutase (SOD), malondialdehyde (MDA), and total antioxidant capacity (T-AOC) were carried out using the commercial kits (Nanjing Jiancheng Bioengineering Institute, Nanjing, China). Alanine aminotransferase (ALT), aspartate aminotransferase (AST), alkaline phosphate (ALP), total protein (TP), albumin (ALB), uric acid (UA), creatinine (CRE), glucose (GLU), and total bilirubin (TBIL) contents were determined with a biochemical analyzer (KHB-ZY 1280, Shanghai Kehua Bio-engineering Co., Ltd, Shanghai, China). The assessment of lysozyme (LZM) activity was conducted by the immunoturbidimetry (ITM) method, and *micrococcus lysodeikticusis* is used as a substrate.

### 2.5. Carcass and Meat Quality

At 42 d of age, two birds randomly selected from each pen were slaughtered, and breast meat yield (percentage of carcass), leg meat yield (percentage of carcass), leg weight, and abdominal fat were documented. Meat quality parameters, including muscle pH, color, water capacity, and shear force, were determined by slaughtering one bird from each pen. Muscle pH was determined by an electronic pH meter at 45 min and 24 h post-mortem. Meat color was determined in duplicate by a Chroma meter and described in CIE-Lab trichromatic system as lightness (L*), redness (a*), and yellowness (b*) at 24 h of slaughtering. Water holding capacity is expressed as drip loss. Shear force was determined at 96 h post-mortem by water-bathing a 20 g muscle sample in a zip-sealed plastic bag at 85 °C for 20 min, on cooling down, shear force values were calculated in triplicate as described by [28].

### 2.6. Statistical Analysis

Data regarding growth performance, hematological characteristics, carcass, and meat quality were analyzed using SPSS software (IBM Corp. IBM SPSS Statistics for Windows, Version 23.0. Armonk, NY, USA). Linear and quadratic effect of the dried MWM were assessed to compare treatment means using regression analysis. Polynomial regression was performed with dried MWM level as a dependent variable when quadratic relationships existed. Comparison between dried 4% MWM and fresh 10.48% MWM were analyzed through independent t-Test. Differences were considered to be statistically significant if probability value less than 0.05, and a trend to difference was set as a probability value less than 0.10. 

## 3. Results

### 3.1. Nutrient Profile of Yellow Mealworm

Nutrient and amino acid composition of fresh mealworm and dried mealworm meal are shown in Table 2. Fresh mealworm contained more than 20% crude protein. Dried mealworm meal had more than 95% dry matter, 50% CP, and 30% crude fat. The lysine, methionine, tryptophan, and threonine content in dried MWM are comparable to soybean meal.

### 3.2. Growth performance

Dietary dried MWM linearly and quadratically increased the starter BW and ADG (*p* < 0.05); FCR improved linear (*p* < 0.05) and quadratically (*p* < 0.10) in the starter phase (Table 3). No linear or quadratic effect of dried MWM on growth performance was observed in the grower phase and the entire growth phase. The t-Test showed that dried 4% MWM had significantly higher BW and ADG and FCR compared to 10.48% fresh mealworm in the starter phase (*p* < 0.05).

### 3.3. Hematological Characteristics

Dried MWM did not significantly linear and quadratic affected hematological characteristics except ALT in the grower phase (Table 4). A diet containing 4% dried MWM reduced the Blood TP (quadratic, *p* = 0.069); however, it increased the MDA level (quadratic, *p* = 0.075) in the starter phase. A higher level of blood UA (linear, *p* = 0.067) was observed in the broiler chicks fed on the diet containing 8% dried MWM in the grower phase. T-Test showed that a diet containing 10.48% fresh mealworm significantly reduced the LZM compared to 4% dried MWM in the grower phase (*p* < 0.05). Broiler chicks fed on the diet containing 10.48% fresh mealworm had reduced T-AOC level (*p* = 0.067) in the starter phase compared to the 4% dried MWM group; however, broiler chicks from the 4% dried MWM group reduced the ALT (*p* = 0.056) and CRE (*p* = 0.051) in the grower phase compared to the 10.48% fresh mealworm group.

### 3.4. Carcass and Meat quality

Dried MWM did not linearly or quadratically affect the carcass composition on d 42 (Table 5). Breast meat percentage, thigh percentage, and abdominal fat percentage are similar among all the dried MWM treatments. The t-Test showed that the diet containing 10.48% fresh mealworm significantly reduced the abdominal fat percentage compared to a diet containing 4% dried MWM. There is no significant difference between the diets 4% dried MWM and 10.48% fresh mealworm regarding the carcass quality of broiler chicks, except for abdominal fat percentage. MWM did not linearly or quadratically affect the meat quality (Table 6). Broiler chicks from the 4% MWM group had a higher a* value (quadratic, *p* = 0.055) 24 h postmortem. The 4% dried MWM diet and the 10.48% fresh mealworm diet did not differ on the meat quality of broiler chicks.

## 4. Discussion

MWM contains a higher amount of CP compared to conventional protein sources. The nutrients and amino acids results showed that the MWM is comparable to fish and soybean meal [29,30]. All nutrients and amino acids of fresh and dried MWM were similar if adjusted to the same dry matter basis. Similar results with slight changes were reported by [12,14,15]. The difference in nutrient profile could be related to induced or natural conditions involved in the production or processing of insects [31]. 

In the current study, dried MWM showed a linear and quadratic effect on starter BW and ADG and partial improvement of FCR during the starter phase. Based on the quadratic regression equation, the predicted dried MWM levels for optimal starter BW and ADG were 4.13% and 3.84%, respectively. The diet containing 4% MWM significantly improved the BW, ADG, and FCR compared to 10.48% fresh mealworm in the starter phase. Similar findings were reported for broiler chicks [15,32], and free-range chicks [33] fed a diet containing MWM. Inconsistent findings showed significantly higher growth performance of broiler chicks [3,12,13,20,23,34], barbary partridges [14], and Japanese quails [11] fed a MWM diet. In our study, MWM did not significantly improve growth performance which could be because of either chitin or polyunsaturated fatty acid (PUFA) in MWM. The exoskeleton of the insect is composed of chitin. Chicken produces chitinase in proventriculus and hepatocytes; however, the digestibility of chitin is limited [19]. Chitin reduces nutrient digestibility [32] and a PUFA-enriched diet reduced the growth performance of broiler chicks [35]. As a promising feedstuff, MWM lacks the nutrient database including digestible AA profile, and the protein and AA profile of MWM varied with the processing method and the growing substrates. In most of the studies with MWM, the diets were formulated based on total amino acid mode, which is far from optimum for broiler chicks. The responses of broilers to a MWM diet formulated on digestible AA profile basis warrants further investigations.

Hematological characteristics of broiler chicks are within the normal physiological ranges [36]. In the grower phase, the linear and quadratic effect of dried MWM was observed on ALT activity. ALT is involved in glucogenesis and amino acid metabolism as well accelerating the metabolism of glucose and proteins [37]. The slight increase of ALT activity within the physiological range, and the unchanged AST activity in the 8% dried MWM diet did not support the concept that MWM hurt the liver function. ALT increase would be partially associated with the increasing level of chloride (~0.4%) in the 8% dried MWM diet. Broiler chicks given 1% salt in drinking water had significantly higher blood ALT [38]. It is a pity that the current study did not consider the dietary electrolytic balance. However, the unchanged growth performance inferred that broiler chicks could tolerate 8% dried MWM. More investigations are needed to further check the dose–response of broilers to the MWM diet fortified with exogenous sodium under the dEB condition. TP is the indicator of the chicken’s health and metabolic changes [39]. TP level changes with nutrition [39]. The reduced TP level in our study could be because of the difference in the inclusion level of MWM in the diet, and a higher level of MWM decreased the blood TP level. MDA and T-AOC are the indicator of oxidative stress [40,41], and CRE is the indicator of kidney functioning [42]. Lipid peroxidation causes fluctuation in the MDA, T-AOC, and CRE level [43,44,45]. The higher level of MDA and lower T-AOC and CRE levels in our study could be due to the presence of a higher quantity of fatty acids in the MWM. A higher level of blood UA in broiler chicks fed the dried 8% MWM diet was observed in the current study. Excessive amino acids in the diet resulted in higher UA [46]. The elevated UA level in our study could be because of the higher quantity of amino acids in MWM; however, chickens can tolerate a higher level of UA [47]. In the grower phase, Broiler chicks fed 10.48% fresh mealworm had a reduced LZM level compared to the counterparts fed 4% dried MWM; however, [48] reported a significantly higher level of LZM in the blood of broiler chicks fed on a diet with 8% housefly maggot meal in the starter phase. LZM is a protein with anti-bacterial characteristics engaged in the defense process of the body [49]. Our study suggests that the lower level of LZM could be because 10.48% fresh MWM could be too high for the broiler chicks. Dietary MWM did not influence the TBIL, AST, ALB, GLU, and SOD. Similar findings were reported in broiler chicks [3,15] and free-range chickens [33] fed MWM. The elevated hematological results of our studies propose that MWM is enriched in PUFA, and a higher proportion of MWM in diet negatively affects the hematological characteristics. Furthermore, a higher level of MWM in the diet reduced the digestibility of broiler chicks; it could be the reason for the lowered growth performance of broiler chicks. The hematological results of our study are contradicted by [2,4] who reported that insects are healthy food; however, a higher level of MWM compromised chicks’ immunity in our experiment. 

Carcass composition was not altered with the addition of MWM, which indicated that MWM did not disturb the carcass characteristics. Abdominal fat percentage was significantly reduced by the diet containing 10.48% fresh mealworm compared to 4% dried MWM. This is in agreement with [50], who reported decreased abdominal fat with increased inclusion level of housefly maggot meal. The carcass quality results of our study are in agreement with [3,33,51], who reported no effect of MWM on carcass quality of broiler chicks and free-range chicks. Inconsistent studies also reported a significant difference in carcass quality of barbary partridges [14], Japanese quails [11] fed MWM diet.

MWM did not significantly affect meat quality. There was no linear or quadratic effect of dried MWM on the meat quality. In our study 4% dried MWM showed elevated a* values, which could be because of muscle pH. Variation in pH changes the meat color [52]. No significant effect of dried or fresh mealworm on meat quality was observed in our study. Meat quality results are in agreement with [12,13,51], who reported no effect of MWM on meat quality. However, [11] reported significantly improved meat quality of Japanese quails due to MWM. No influence in the water holding capacity was observed in the current study, which was inconsistent with the results of [21], who reported improved water holding capacity by MWM. The diet with dried MWM has a pleasant smell, like frying oil. During the de-feathering process with hot water before carcass composition determination, the flavor of ’Luzhu Tofu’ (Chinese food) could be smelt in broiler chicks fed on 4% and 8% dried MWM diets. Thus, the functional ingredients in the MWM and the flavor of the meat are worthy of being investigated. Flavor and fatty acid profile of meat warrant further study in future research using MWM as protein source or functional additive in broiler chicks.

In the current study, the efficacy of dried mealworm meal and fresh mealworm was compared. Fresh mealworm induced a retarded growth rate and inferior feed efficiency during the starter phase. However, the compensatory growth in the grower phase led to comparable growth performance during the entire phase. These results demonstrated that the broiler might not accustom to the diet containing fresh meal at an early age. Although the blood LZM level was compromised, the abdominal fat decreased at the grower phase in the fresh mealworm group compared to the dried mealworm meal group. Together with the unchanged blood profile and meat quality parameters, the overall results attested that both the dried and fresh mealworm were of about the same cumulative efficiency in broilers and can be safely included in the broiler diet.

## 5. Conclusions

The present study suggests that yellow mealworm could be an alternative protein feedstuff for the broiler diet. Dietary inclusion of dried yellow mealworm meal at a 4% dosage had the potential to promote growth performance, especially in the starter phase. Both dried and fresh yellow mealworm is acceptable and had a similar effect. Further research is needed to investigate the effect of dried yellow mealworm on broiler meat quality.

## Figures and Tables

**Table 1 animals-10-00224-t001:** Ingredients and nutritional level of the diets.

Ingredients (% as Fed Basis)	Mealworm Inclusion							
	Starter Phase				Grower Phase			
	0%	2%	4% ^a^	8%	0%	2%	4% ^a^	8%
Corn	56.5	58	59.5	62.5	59.7	61.2	62.8	65.8
Soybean meal (43%)	31.5	28.7	25.9	20.3	26.2	23.4	20.6	15
Yellow mealworm	0	2	4	8	0	2	4	8
Rapeseed meal	3	3	3	3	4	4	4	4
Cottonseed meal	2	2	2	2	2.5	2.5	2.5	2.5
Vegetable oil	2.90	2.15	1.42	0	3.96	3.22	2.46	0.97
Di-Calcium phosphate	1.81	1.74	1.68	1.54	1.51	1.45	1.38	1.25
Limestone	1.28	1.33	1.39	1.49	1.21	1.26	1.31	1.42
Salt	0.35	0.35	0.35	0.35	0.35	0.35	0.35	0.35
DL-methionine	0.21	0.21	0.21	0.22	0.12	0.12	0.12	0.12
L-lysine HCL	0.14	0.16	0.18	0.23	0.14	0.16	0.19	0.24
Threonine	0.03	0.03	0.03	0.04	0.02	0.02	0.03	0.03
Tryptophan								0.01
Vitamin Premix ^①^	0.02	0.02	0.02	0.02	0.02	0.02	0.02	0.02
Mineral Premix^②^	0.2	0.2	0.2	0.2	0.2	0.2	0.2	0.2
Choline chloride (50 %)	0.1	0.1	0.1	0.1	0.1	0.1	0.1	0.1
Nutrient Profile ^③^								
Apparent metabolizable energy (kcal/kg)	2950	2950	2950	2952	3050	3050	3050	3050
Crude protein, g/kg	207	208	208	209	193	195	196	197
Calcium, g/kg	10	10	10	10	9	9	9	9
Total phosphorus, g/kg	7.1	7.1	7.0	6.8	6.5	6.5	6.3	6.1
Available phosphorus, g/kg	4.5	4.5	4.5	4.5	4	4	4	4
Lysine, g/kg	11.28	11.45	11.56	11.62	10.53	10.71	10.88	10.95
Methionine, g/kg	4.55	4.66	4.59	4.73	3.59	3.67	3.58	3.62
Methionine + cysteine, g/kg	7.63	7.85	7.78	7.84	6.93	7.05	6.89	6.97
Tryptophan, g/kg	2.45	2.28	2.24	2.12	2.13	2.05	1.97	1.95
Threonine, g/kg	6.98	6.85	6.92	7.01	6.59	6.53	6.49	6.58
dEB (mEq/kg) ^④^	209.03	196.65	184.32	158.88	188.95	176.55	163.47	135.70

**^a^** Feed formulation for fresh (10.48%) mealworm corresponded to that of dried 4% mealworm meal (MWM) except for the type of mealworm; ^①^ The vitamin premix supplied the following per kg of complete feed: vitamin A, 12,500 IU; vitamin D_3_, 2500 IU; vitamin K_3_, 2.65 mg; vitamin B_1_, 2 mg; vitamin B_2_, 6 mg; vitamin B_12_, 0.025 mg; vitamin E, 50 IU; biotin, 0.0325 mg; folic acid, 1.25 mg; pantothenic acid, 12 mg; niacin, 50 mg; ^②^ The mineral premix supplied the following per kg of complete feed: Cu, 8 mg; Zn, 75 mg; Fe, 80 mg; Mn, 100 mg; Se, 0.15 mg; I, 0·35 mg; ^③^ On 88% dry matter basis, the AME value was calculated, others were analyzed values; ^④^ dEB (dietary electrolyte balance) = Na+K−Cl.

**Table 2 animals-10-00224-t002:** Nutrient and amino acids composition of the yellow mealworm.

Items	Fresh Mealworm	Dried Mealworm Meal
Dry matter (%)	36.41	95.40
Crude protein (%)	20.15	52.89
Crude fat (%)	11.49	30.05
Calcium (%)	0.096	0.25
Phosphorus (%)	0.28	0.74
Gross energy (kcal/kg)	2132	5586
Amino Acids (%)		
Aspartic acid	1.59	4.20
Threonine	0.77	2.01
Serine	0.85	2.26
Glutamic acid	2.37	6.2
Glycine	1.02	2.73
Alanine	1.49	3.9
Cysteine	0.27	0.70
Valine	1.13	2.99
Methionine	0.25	0.63
Isoleucine	0.88	2.30
Leucine	1.46	3.85
Tyrosine	1.32	3.44
Phenylalanine	0.68	1.77
Histidine	1.10	2.91
Lysine	1.08	2.8
Arginine	1.03	2.73
Proline	0.59	1.62
Tryptophan	0.14	0.36

**Table 3 animals-10-00224-t003:** Growth performance of the broiler chicks fed on the MWM diet.

Items ^a^	Mealworm Meal Inclusion					*P*-Value		
	0%	2%	4%	8%	10.48%	Linear ^b^	Quadratic ^b^	t-Test ^c^
Starter								
BW (g)	619 ± 12.8	605 ± 11.5	664 ± 16.6	609 ± 19.1	610 ± 11.1	0.033	0.019	0.036
ADG (g)	27.6 ± 0.60	26.9 ± 0.57	29.5 ± 0.82	26.7 ± 0.94	27.2 ± 0.53	0.027	0.016	0.024
ADFI* (g)	39.1 ± 0.70	38.7 ± 0.78	41.5 ± 0.89	40.9 ± 1.08	41.4 ± 0.50	0.310	0.130	0.680
FCR	1.42 ± 0.02	1.44 ± 0.02	1.41 ± 0.02	1.54 ± 0.04	1.53 ± 0.03	0.030	0.055	0.002
Grower								
BW (g)	2213 ± 48.7	2219 ± 43.6	2253 ± 62.3	2219 ± 45.3	2273 ± 29.8	0.861	0.692	0.753
ADG (g)	74.2 ± 2.03	75.4 ± 1.93	74.9 ± 2.37	75.5 ± 2.24	77.5 ± 1.35	0.949	0.778	0.347
ADFI* (g)	163.2 ± 2.73	161.7 ± 3.20	163.8 ± 2.12	161.4 ± 3.79	158.5 ± 1.85	0.840	0.569	0.106
FCR	2.21 ± 0.06	2.15 ± 0.03	2.20 ± 0.06	2.15 ±0.06	2.05 ± 0.04	0.668	0.374	0.143
Entire feeding period								
ADG (g)	48.8 ± 1.04	48.8 ± 0.76	50.2 ± 1.38	48.6 ± 1.17	50.2 ± 0.78	0.550	0.386	0.898
ADFI*(g)	95.6 ± 1.4	94.2 ± 1.48	97.2 ± 1.15	94.8 ± 1.71	95.0 ± 0.82	0.343	0.157	0.102
FCR	1.96 ± 0.03	1.93 ± 0.02	1.95 ± 0.04	1.96 ± 0.03	1.89 ± 0.02	0.816	0.757	0.441

* Feed intake was adjusted to 88% dry matter basis; ^a^ BW, body weight; ADG, average daily gain; ADFI, average daily feed intake; FCR, feed conversion ratio (feed: gain, g:g); ^b^ Linear and quadratic effect of dried MWM supplementation were evaluated using regression analysis; ^c^ t-test of 4% dried MWM and 10.48% fresh mealworm.

**Table 4 animals-10-00224-t004:** Hematological characteristics of broiler chicks fed on MWM diet.

Items ^a^	Mealworm Meal Inclusion					*P*-Value		
	0%	2%	4%	8%	10.48%	Linear ^b^	Quadratic ^b^	t-Test ^c^
Starter								
TBIL (μ mol/L)	3.60 ± 0.341	3.59 ± 0.375	4.47 ± 0.582	3.64 ± 0.255	3.40 ± 0.275	0.265	0.183	0.240
AST (U/L)	245 ± 9.47	250 ± 16.9	251 ± 12.4	255 ± 14.8	259 ± 11.4	0.997	0.942	0.664
ALB (g/L)	15.5 ± 0.42	15.4 ± 0.56	14.3 ± 0.62	14.6 ± 0.55	14.7 ± 0.54	0.585	0.336	0.535
UA (μ mol/L)	359 ± 69.1	547 ± 121	355 ± 73.4	472 ± 104	389 ± 94.8	0.295	0.130	0.642
ALT (U/L)	2.2 ± 0.19	2.1 ± 0.22	1.6 ± 0.25	1.4 ± 0.21	2.0 ± 0.47	0.779	0.503	0.347
TP (g/L)	34.5 ± 1.03	34.7 ± 1.35	30.3 ± 1.04	31.7 ± 1.21	31.6 ± 1.43	0.165	0.069	0.379
GLU (mmol/L)	13.8 ± 0.62	12.4 ± 0.85	12.0 ± 0.85	11.2 ± 0.96	11.0 ± 0.94	0.857	0.685	0.827
CRE (μ mol/L)	58.1 ± 5.15	57.6 ± 4.08	52.4 ± 3.66	55.7 ± 2.76	53.6 ± 3.63	0.708	0.486	0.213
ALP (×10^3^U/L)	261 ± 31.5	172 ± 39.6	258 ± 57.3	148 ± 27.5	189 ± 30.6	0.165	0.061	0.357
LZM (μ g/mL)	0.72 ± 0.05	0.75 ± 0.07	0.68 ± 0.05	0.68 ± 0.08	0.47 ± 0.05	0.826	0.587	0.108
T-AOC (U/mL)	1.01 ± 0.07	0.86 ± 0.12	1.29 ± 0.26	1.30 ± 0.29	0.63 ± 0.12	0.560	0.316	0.067
MDA (nmol/mL)	2.95 ± 0.17	3.01 ± 0.21	3.59 ± 0.14	3.14 ± 0.19	3.79 ± 0.21	0.088	0.075	0.267
SOD (U/mL)	51.7 ± 6.2	42.6 ± 3.58	52.2 ± 4.44	49.4 ± 3.46	53.0 ± 3.9	0.295	0.156	0.727
Grower								
TBIL (μ mol/L)	3.44 ± 0.22	3.33 ± 0.36	3.84 ± 0.39	4.83 ± 0.56	2.75 ± 0.24	0.440	0.942	0.118
AST (U/L)	377 ± 31.1	389 ± 28	375 ± 19.9	461 ± 19.1	375 ± 18.7	0.229	0.294	0.676
ALB (g/L)	15.3 ± 0.41	15.2 ± 0.29	15.1 ± 0.57	16.9 ± 0.72	15.6 ± 0.5	0.213	0.488	0.249
UA (μ mol/L)	429 ± 50	471 ± 79.4	347 ± 62.8	890 ± 218	631.0 ± 162	0.067	0.159	0.172
ALT (U/L)	4.7 ± 0.32	5.1 ± 0.87	3.5 ± 0.43	5.9 ± 0.57	5.1 ± 0.39	0.035	0.036	0.056
TP (g/L)	36.0 ± 1.18	33.8 ± 1.07	35.2 ± 1.29	38.7 ± 1.78	35.7 ± 1.52	0.151	0.804	0.346
GLU (mmol/L)	11.7 ± 0.46	11.6 ± 0.52	13.2 ± 0.48	12.3 ± 0.59	12.8 ± 0.4	0.191	0.095	0.528
CRE (μ mol/L)	55.9 ± 2.92	56.0 ± 1.64	51.9 ± 2.44	56.7 ± 3.54	53.5 ± 2.39	0.434	0.315	0.051
ALP (×10^3^U/L)	318 ± 39.7	228 ± 33.4	323 ± 75.5	225 ± 48.5	293 ± 65.8	0.310	0.129	0.535
LZM (μ g/mL)	0.57 ± 0.1	0.51 ± 0.15	0.74 ± 0.15	0.63 ± 0.22	0.35 ± 0.06	0.712	0.419	0.027
T-AOC (U/mL)	11.9± 0.63	13.1 ± 0.94	13.0 ± 1.32	17.1 ± 2.57	13.8 ± 1.6	0.527	0.470	0.698
MDA (nmol/mL)	3.45 ± 0.12	3.83 ± 0.24	4.86 ± 0.28	4.35 ± 0.61	4.27 ± 0.26	0.241	0.211	0.443
SOD (U/mL)	82.6 ± 9.92	98.3 ± 8.82	101 ± 7.32	103 ± 9.13	106 ± 8.12	0.743	0.747	0.725

^a^ TBIL, total superoxide dismutase; AST, aspartate aminotransferase; ALB, albumin; UA, uric acid; ALT, alanine aminotransferase; TP, total protein; GLU, glucose; CRE, creatinine; ALP, alkaline phosphate; LZM, lysozyme; T-AOC, total antioxidant capacity; MDA, malondialdehyde; SOD, total superoxide dismutase; ^b^ Linear and quadratic effect of dried MWM supplementation were evaluated using regression analysis; ^c^ t-test of 4% dried MWM and 10.48% fresh mealworm.

**Table 5 animals-10-00224-t005:** Carcass quality of the broiler chicks fed on the MWM diet.

Items	Mealworm Meal Inclusion					*P*-Value		
	0%	2%	4%	8%	10.48%	Linear ^a^	Quadratic ^a^	t-Test ^b^
Starter								
Spleen (%)	1.15 ± 0.1	0.95 ± 0.07	1.01 ± 0.11	1.17 ± 0.14	1.13 ± 0.06	0.280	0.771	0.647
Thymus (%)	2.19 ± 0.16	2.24 ± 0.24	2.76 ± 0.28	2.44 ± 0.26	2.14 ± 0.22	0.384	0.246	0.285
Bursa (%)	1.94 ± 0.14	1.95 ± 0.17	1.87 ± 0.14	1.97 ± 0.19	1.64 ± 0.12	0.908	0.733	0.224
Grower								
Chick BW (g)	2219 ± 43.7	2201 ± 30.5	2217 ± 60.4	2283 ± 41.8	2261 ± 26.8	0.684	0.946	0.507
Avg. Carcass wt. (g)	2032 ± 43.4	1990 ± 35.2	2031 ± 56.5	2029 ± 44.4	2034 ± 31.3	0.777	0.565	0.936
Eviscerated wt. (g)	1644 ± 38.3	1625 ± 38.4	1643 ± 50	1644 ± 35.5	1634 ± 25.1	0.936	0.782	0.898
Dressing (%)	80.8 ± 0.28	80.9 ± 0.5	80.5 ± 0.57	81.2 ± 0.3	80.2 ± 0.36	0.612	0.491	0.735
Breast (%)	23.4 ± 0.77	23.9 ± 0.43	23.5 ± 0.67	24.7 ± 0.48	24.7 ± 0.62	0.579	0.389	0.303
Thigh (%)	20.7 ± 0.53	21 ± 0.41	20.6 ± 0.83	20.5 ± 0.47	20.5 ± 0.35	0.907	0.769	0.907
Abdominal fat (%)	1.97 ± 0.13	2.01 ± 0.08	2.11 ± 0.12	1.98 ± 0.10	1.92 ± 0.12	0.648	0.566	0.032
Spleen (%)	1.17 ± 0.11	1.07 ± 0.12	1.03 ± 0.05	1.06 ± 0.08	1.08 ± 0.08	0.847	0.990	0.765
Thymus (%)	1.32 ± 0.14	1.44 ± 0.14	1.34 ± 0.09	1.7 ± 0.10	1.72 ± 0.15	0.314	0.234	0.077
Bursa (%)	0.47 ± 0.03	0.47 ± 0.03	0.44 ± 0.03	0.47 ± 0.02	0.53 ± 0.03	0.657	0.453	0.101

^a^ Linear and quadratic effect of dried MWM supplementation were evaluated using regression analysis; ^b^ t-test of 4% dried MWM and 10.48% fresh mealworm. BW, Body weight; Avg., Average; wt., Weight.

**Table 6 animals-10-00224-t006:** Meat quality of the broiler chicks fed on MWM diet.

Items	Mealworm Meal Inclusion					*P*-Value		
	0%	2%	4%	8%	10.48%	Linear ^a^	Quadratic ^a^	t-Test ^b^
Drip loss (%)	2.38 ± 0.36	2.20 ± 0.16	2.14 ± 0.19	1.96 ± 0.23	2.03 ± 0.25	0.980	0.840	0.601
Cooking loss (%)	25.6 ± 0.43	25.1 ± 0.78	24.5 ± 0.39	23.1 ± 0.47	25.4 ± 0.74	0.875	0.860	0.831
Shear force (N)	18.2 ± 0.85	15.5 ± 0.78	14 ± 0.58	13.7 ± 0.53	17 ± 1.23	0.265	0.973	0.212
45 min								
L^*^	51.7 ± 0.98	52.3 ± 0.51	50.9 ± 0.63	50.6 ± 0.83	51.6 ± 0.57	0.606	0.419	0.684
a^*^	5.89 ± 0.48	5.39 ± 0.58	6.24 ± 0.48	4.86 ± 0.35	5.59 ± 0.28	0.210	0.123	0.399
b^*^	10.3 ± 0.47	11.8 ± 0.51	11.8 ± 0.47	12.3 ± 0.76	10.9 ± 0.42	0.536	0.422	0.259
pH	6.03 ± 0.05	6.13 ± 0.04	6.16 ± 0.06	6.14 ± 0.03	6.08 ± 0.05	0.485	0.848	0.304
24 h								
L^*^	55.6 ± 1.38	56 ± 0.8	53.5 ± 0.95	53.8 ± 1.05	54.9 ± 0.83	0.522	0.258	0.587
a^*^	5.60 ± 0.62	5.32 ± 0.62	7.03 ± 0.54	5.45 ± 0.48	5.37 ± 0.4	0.091	0.055	0.134
b^*^	12.4 ± 0.58	13.6 ± 0.46	12.4 ± 0.52	12 ± 0.49	11.5 ± 0.75	0.147	0.207	0.196
pH	5.25 ± 0.05	5.21 ± 0.06	5.36 ± 0.06	5.31 ± 0.07	5.24 ± 0.07	0.451	0.210	0.259

L*, lightness; a*, redness; b*, yellowness; ^a^ Linear and quadratic effect of dried MWM supplementation were evaluated using regression analysis; ^b^ t-test of 4% dried MWM and 10.48% fresh mealworm.

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
