# Peer review of "Evaluation of Yellow Mealworm Meal as a Protein Feedstuff in the Diet of Broiler Chicks"

_animals, 2020, doi:10.3390/ani10020224_

Round 1
Reviewer 1 Report
This article aims to evaluate the effect of 1) increasing doses of mealworm meal and 2) drying the mealworm on growth performance, hematological characteristics and carcass and meat quality of broiler chicks.
In my opinion the experimental design is approppriate but my major concern is the statistical analysis:
1) To examine incremental additions of a nutrient to a diet in order to identify an optimal level and/or response surface, multiple comparison tests are unlikely to be appropriate and polynomial contrasts are generally the most effective statistical tests. Data should be reanalysed and presented in tables.
2) A multiple comparison test can be used to assess the effect of drying
Therefore, my recommendation is to reconsider after major revision.
Other comments
Line 15. Please reword
Line 18, 62 and 63. Define the abbreviation the first time you use it.
Line 26-28. Please reword
Line 28-29. Is it not contradictory? Please reword
Introduction
Authors use fresh meal as well. The hypothesis for using fresh meal should be clearly stated in the introduction and objectives.
Line 48. Add reference
Line 73-75. Is there a hypothesis of why defining dosage is so important? Do you expect to find a quadratic effect? Auhtors need to provide more information on this subject as this is the novelty of this work.
Line 75-77. Reword the objective because the novelty of this article is the dosage.
Line 84. What did you use to dry? Provide information on the device you used and conditions of the process (duration).
Line 86. Please reword. How did you store fresh samples until analysis or the feeding trial?
Line 86-87. Ok, but information on the methods used for the analysis are missing. Moreover, did you analyse diets?
Line 89-91. Provide information on the dimensions of the pen. Were pens provided with individual feeders? Was feed offered ad libitum?
Line 98-99. Is it on a calculated basis or did authors actually determine the energy, CP...? Indicate the form of the feed in each phase.
Table 1. Ingredients instead of feed formula. Indicate whether ingrediets are as fed or on DM basis. Provide information in g/kg instead of %. Define abbreviations at table footnote. Superscript 3 is missing
Line 110-112. Did you measure mortality?. Define documented data, this is very poor. Indicate when you measured feed intake and how.
Line 126-134. I understand that 3 birds were slaughtered, were they randomly selected?
Line 136-145. As I have stated before I do not consider multiple comparison tests such as Duncan's appropriate to study the response effect of any variable to increasing doses of a certain treatment. Authors need to reanalyze data using polynomial contrast to see whether there is a linear or a quadratic response to increasing doses. Until this is done there is no sense in revising the article beyond material and methods section.
Author Response
Dear Editors and Reviewers,
Thank you very much for your precious comments and suggestions about our manuscript submitted to your journal. Your suggestions gave us much to learn and helped improve our scientific writing to a great extent. We have revised the manuscript according to the comments and suggestions, and the amendments are highlighted with yellow in the revised manuscript. Below you will find our point-by-point responses to your comments and questions. The whole manuscript has been carefully rechecked by ourselves. We do hope we could understand your questions correctly and have given the right answers in the revised manuscript. Please feel free to inform me if there are still some questions. Thank you very much in advance!
Best wishes and kind regards,
Dr. Guang-Hai Qi
Professor of Poultry Nutrition
Feed Research Institute
Chinese Academy of Agricultural Sciences
E-mail: qiguanghai@caas.cn

Reviewer 2 Report
The content of the paper is interesting and fits the Animals scope but I see some problems in this paper necessary to correct or explain in details:
Introduction
In my opinion, the introduction section is too long, especially the part from L37 to 57. Please justify 10.48% treatment, why did you use fresh worms?Materials and methods:
You need to provide an average initial body weight of birds What about the sex of the birds? Please provide information about diet preparation, a form of the diet, etc. What about feed 10.48% composition I don’t see it in table 1 Please provide Cl content in diets or calculate DEB for all diets. In my opinion, your diets are different in DEB. I don’t see an explanation for superscript “3” in table 1 and a standarise number of decimals Please provide the determined nutritional value of diets – it is crucial in that type of study Please provide in details how performance parameters were calculated L114: males or females,? L126: males or females?Statistical analyses
If the aim of the study was to asses optimum inclusion ratio you should use linear contrasts – it is classic dose trialResults and discussion section:
In my opinion, authors should use linear contrasts to describe and results Please put more attention to explain why WMW affected performance parameters – try to explain it Keep in mind that you use total AA (instead of digestible values) to formulae diets, it is far from optimum for those birds. Try to discuss your results keeping in mind the above comment.Author Response
Dear Editors and Reviewers,
Thank you very much for your precious comments and suggestions about our manuscript submitted to your journal. Your suggestions gave us much to learn and helped improve our scientific writing to a great extent. We have revised the manuscript according to the comments and suggestions, and the amendments are highlighted with yellow in the revised manuscript. Below you will find our point-by-point responses to your comments and questions. The whole manuscript has been carefully rechecked by ourselves. We do hope we could understand your questions correctly and have given the right answers in the revised manuscript. Please feel free to inform me if there are still some questions. Thank you very much in advance!
Best wishes and kind regards,
Dr. Guang-Hai Qi
Professor of Poultry Nutrition
Feed Research Institute
Chinese Academy of Agricultural Sciences
E-mail: qiguanghai@caas.cn

Round 2
Reviewer 1 Report
The manuscript has been improved to a certain extent but still requires important improvements before I can consider it acceptable.
Line 26-27. Please be more specific. What variable responded quadratically and which one linearly.
Line 29-30. According to the discussion this was not an effect of MWN, but rather of Na. Please either delete or reword
Line 79-81. Please reword the objectives to clearly state the two objectives of the trial.
Line 89-90. Check English
Line 92-95. Please provide the identification number of each AOAC procedure used.
Line 96. Were birds vaccinated?
Line 104-105. How was the fresh meal fed? formulated in the concentrate? If so, information is missing in Table 1. Please be more specific.
Line 107-108. Please delete. This is a result reported in Table 1.
Table 1. Are ingredients on DM basis or as fed? Is nutrient profile on DM basis or as fed? If as fed, provide DM content. Delete "dried". Provide only 1 decimal for corn and soybean meal. Provide two decimals for all the others. "Nutrient profile" instead of Nutrient level. When actual values are available, provide just those. When not available, indicate that are calculated.
Line 129. For fresh worm meal, did you use adjusted to 88% DM FI?
Line 155-158. Statistical analysis on the comparison between dried and fresh worm meal is missing. It is not clear what authors have compared. Authors need to keep a coherence with the objectives. That is why they need to be reworded to clearly state what the authors were aiming at. Was it not to assess the effect of drying? If so, what is the sense of comparing fresh meal vs control or 2% dry meal?
Line 159-166. Delete
Line 169-170. Please delete. This is a result.
Line 170-171. Please delete. This is discussion.
Line 176-181 and elsewhere. Only a significant quadratic effect could lead to a statement such as: ‘values were highest/lowest at the intermediate addition level’, whereas significant linear and quadratic effects could lead to a statement such as: ‘values increased/decreased at an increasing/decreasing rate’.
Line 178-179 and elsewhere in the manuscript. Not reported on tables. If authors perform a polynomial analysis there is no need for a multiple comparison test (because it is not appropriate). Authors need to keep the coherence with the objective.
Table 3. "starter" instead of "21 d" and "grower" instead of "42d". "entire feeding period" instead of "1-42d". Delete decimals for BW, 1 decimal for ADG and ADFI, and two decimals for FCR. Provide SEM with one decimal more than that reported with means. Delete superscript c: not necessary.
Table 4 and elsewhere. "starter" instead of "21 d" and "grower" instead of "42d".
Discussion needs to be greatly improved. Reword discussion avoiding repeating results or using P values. In addition, it might change depending on the results obtained after the new statistical analysis. For instance, if one of the objectives is to assess the effect of drying, this discussion is missing.
Line 217. compared to what?
Line 217-218 Provide a reference
Line 254. Is it not lower?
Line 254-256. Are the other values beyond the maximum healthy threshold?
Conclusions
Line 292-293 Please reword, authors cannot refer to a specif dose
Line 293-294. On what grounds?
Line 294-295. On what grounds?
Author Response
Dear Reviewer,
Thank you very much for your precious comments and suggestions about our manuscript submitted to your journal. Your suggestions gave us much to learn and helped improve our scientific writing to a great extent. We have revised the manuscript according to the comments and suggestions, and the amendments are highlighted with yellow in the revised manuscript. Below you will find our point-by-point responses to your comments and questions. The whole manuscript has been carefully rechecked by ourselves. We do hope we could understand your questions correctly and have given the right answers in the revised manuscript. Please feel free to inform me if there are still some questions. Thank you very much in advance!
Best wishes and kind regards,
Dr. Guang-hai Qi
Professor of Poultry Nutrition
Feed Research Institute
Chinese Academy of Agricultural Sciences
E-mail: qiguanghai@caas.cn

Round 3
Reviewer 1 Report
The manuscript has been greatly improved, but there are still a few changes to address before I can consider it acceptable for publication. Therefore, my decision is to accept the manuscript after a minor revision.
It is often very wise to state clearly in the statistical analysis section the probability levels that are being accepted as representative of statistical differences and, possibly, trends to differences (for the T test), and be consistent with this criteria. Then, try not to treat numerical differences as being equivalent to statistical differences as in Line 183 and 299-301.
Author Response
Dear Reviewer,
Thank you very much for your valuable suggestions about our manuscript submitted to your journal. We had updated our manuscript according to your worthy advice, recommendations, and suggestions. Your suggestions gave us much to learn and helped improve our scientific writing to a great extent, especially statistical analysis. The whole manuscript has been carefully rechecked by ourselves. Please feel free to inform me if there are still some questions. Thank you very much in advance!
Best wishes and kind regards,
Dr. Guang-hai Qi
Professor of Poultry Nutrition
Feed Research Institute
Chinese Academy of Agricultural Sciences
E-mail: qiguanghai@caas.cn
